# Comparative dissection of the peripheral olfactory system of the Chagas disease vectors *Rhodnius prolixus* and *Rhodnius brethesi*

Florencia Campetella[1], Rickard Ignell[2], Rolf Beutel[3], Bill S. Hansson[1], Silke Sachse[1]*

1 Department of Evolutionary Neuroethology, Max Planck Institute for Chemical Ecology, Jena, Germany, 2 Unit of Chemical Ecology, Department of Plant Protection Biology, Swedish University of Agricultural Sciences, Alnarp, Sweden, 3 Institute for Zoology and Evolutionary Biology, Friedrich Schiller University, Jena, Germany

* ssachse@ice.mpg.de

**Data Availability Statement:** All relevant data are within the manuscript and its Supporting Information files.

## Abstract

American trypanosomiasis, or Chagas disease, is transmitted by both domestic and sylvatic species of Triatominae which use sensory cues to locate their vertebrate hosts. Among them, odorants have been shown to play a key role. Previous work revealed morphological differences in the sensory apparatus of different species of Triatomines, but to date a comparative functional study of the olfactory system is lacking. After examining the antennal sensilla with scanning electronic microscopy (SEM), we compared olfactory responses of *Rhodnius prolixus* and the sylvatic *Rhodnius brethesi* using an electrophysiological approach. In electroantennogram (EAG) recordings, we first showed that the antenna of *R. prolixus* is highly responsive to carboxylic acids, compounds found in their habitat and the headspace of their vertebrate hosts. We then compared responses from olfactory sensory neurons (OSNs) housed in the grooved peg sensilla of both species, as these are tuned to these compounds using single-sensillum recordings (SSRs). In *R. prolixus*, the SSR responses revealed a narrower tuning breath than its sylvatic sibling, with the latter showing responses to a broader range of chemical classes. Additionally, we observed significant differences between these two species in their response to particular volatiles, such as amyl acetate and butyryl chloride. In summary, the closely related, but ecologically differentiated *R. prolixus* and *R. brethesi* display distinct differences in their olfactory functions. Considering the ongoing rapid destruction of the natural habitat of sylvatic species and the likely shift towards environments shaped by humans, we expect that our results will contribute to the design of efficient vector control strategies in the future.

## Author summary

An estimated eight million people worldwide are infected with American trypanosomiasis, also known as Chagas disease, whose causative agent is the parasite *Trypanosoma*

**Funding:** This work was funded by the Max Planck Society to F.C., B.S.H., S.S. The funders had no role in study design, data collection and analysis, decision to publish, or preparation of the manuscript.

**Competing interests:** The authors have declared that no competing interests exist.

*cruzi*. Over a hundred species of insects belonging to the Tritatomine subfamily are vectors of the disease, as they spread *T. cruzi* through their feaces. Several studies have highlighted the importance of olfaction for host-seeking behavior in these insects, which enables them to locate their vertebrate hosts and to obtain their vital blood meal. Vector control strategies have been the most efficient policy to combat the spread of Chagas disease by triatomine insects. However, recent changes in the natural habitats of these insects challenge the efficacy of these strategies, as species so far thought to be exclusive to sylvatic environments are now frequently found in peridomestic areas. In this context, understanding how triatomines with different distributions detect odors to locate their hosts and choose their habitats is highly relevant. In this study, we compare the olfactory system of the widely distributed *Rhodnius prolixus* and a sylvatic sibling *Rhodnius brethesi* at a morphological and functional level. We reveal that detection of host and habitat volatiles share many similarities, but also exhibit pronounced differences between species.

## Introduction

Chagas disease or American trypanosomiasis, caused by infection with the protozoan *Trypanosoma cruzi*, is a chronic disease that is endemic in 21 Latin American countries, where it significantly affects the most vulnerable inhabitants. It is estimated that its prevalence in some areas can be as high as 5%, and its annual burden in health care costs sums up to 600 million dollars [1]. Already in 1905 it was shown that blood-sucking insects belonging to the Triatominae subfamily (Heteroptera: Reduviidae) transmit *T. cruzi* through their faeces. To date, the most effective and successful methods to control the spread of Chagas disease have been vector control policies. Wide-spread use of pesticides and training of local communities to identify and kill the insects are the most efficient strategies to date [2]. However, with the appearance of pesticide-resistant insects, new management strategies are urgently needed.

Triatominae is a poorly defined and possibly paraphyletic group of the predaceous true bugs of the family Reduviidae [3]. All 151 described species, phylogenetically grouped into five tribes [4], are capable of transmitting Chagas disease [5]. From these, some species, such as *Rhodnius prolixus* and *Triatoma infestans*, are considered particularly important from an epidemiological standpoint, as they are widely distributed in South America and are often found, though not exclusively, in domiciliated areas [6–10]. However, most of the species of the Rhodniini tribe, to which *R. prolixus* belongs, are thought to have a more restricted distribution in sylvatic areas. Within the areas they inhabit, triatomines often find refuge in palm trees [11] and, depending on the number of palm tree species in which they nest, they can be classified as refuge generalists or specialists [12–14]. While *R. prolixus* is known to be of the first type, an interesting example of a sylvatic specialist species is *Rhodnius brethesi* which, so far, has only been found on the palm tree species *Leopoldina piassaba* [13, 14]. Despite the interesting nature of these associations, studies on sylvatic species has been marginal, with most of the research focused on domiciliated species. However, as deforestation and climate change increase [12, 15], sylvatic species will lose their natural habitats and might find refuge in domestic and peridomestic areas [16], putting its inhabitants at higher risk and becoming a public health problem. Thus, in order to design better vector control strategies, a thorough understanding of the differences and similarities between triatomine species having different habitat requirements is needed.

Being active at night, triatomines make use of physical and chemical cues to find their hosts [17, 18]. Several studies have highlighted the importance of olfaction for host-seeking behavior in these insects [19–24]. Terrestrial vertebrates, the main host for these obligatory haematophagaous insects, emit odor signatures that can be composed of up to 1000 different volatiles

[25–27], many of them being produced by the skin microflora [28]. Previous work has shown that *T. infestans* and *R. prolixus* make use of some of these volatiles to find their hosts [20, 21, 23, 29–33]. In particular, carbon dioxide, 1-octen-3-ol, acetone, several amines, as well as carboxylic acids are attractive cues for *R. prolixus* [21, 34, 35], whose detection is achieved by specialized olfactory sensilla in the antenna [31, 33, 36–39]. However, studies comparing olfactory responses between different triatomine species are lacking.

In insects, differences in olfactory tuning have been reported between species of the same genus, and between wild and domestic insects of the same species [40–44]. In triatomines, previous studies have shown that the number of olfactory sensilla is correlated with the complexity and number of ecotypes in which the insects are found [45, 46]. For instance, domestic species with stable environments have a lower number of chemosensory sensilla than their sylvatic relatives [45–49]. Furthermore, a reduced expression of olfactory binding proteins (OBPs) and chemosensory proteins (CSPs) are found in domiciliated *Triatoma brasiliensis*, compared to sylvatic and peridomestic ones. [50]. In this study, we hypothesized that the morphology and tuning of the olfactory system of *R. prolixus* and *R. brethesi* reflect the different habitat distribution and requirements of the two species. To test this, we used a comparative approach to characterize the peripheral olfactory system of the widely distributed generalist *R. prolixus* and the sylvatic specialist *R. brethesi* at an anatomical and functional level.

## Material and methods

### Insect rearing

Insects were reared as previously described [51, 52]. Adult males of *R. brethesi* and *R. prolixus*, starved for 3–4 weeks, were used in the experiments. Batches of insects were kept in individual boxes with a Light:Dark cycle set to 12:12 h. The boxes were placed inside a chamber at 25˚C and 60% relative humidity. Each insect was used at the beginning of the scotophase, as it has been shown that olfactory acuity is higher at this timepoint [20, 53].

Laboratory rearing has been shown to have a species-specific impact on the number and distribution of olfactory and mechanosensory sensilla [49]. However, according to previous work, in the case of *R. prolixus* this effect is either non-existant or only moderate [54]. While, in *R. brethesi* an increase in the density of mechanosensory sensilla (bristles), and a reduction in the number of trichoid and basiconic sensilla has been observed in laboratory-rerared insects compared to wild ones [48], it was not possible to include specimens of this species from the field.

### SEM

The heads of the insects, including the antennae were fixed with 2.5% (v/v) glutaraldehyde in cacodylate buffer (pH 7.4) for 60 min. Afterwards, the samples were washed three times for 10 min with cacodylate buffer and dehydrated in ascending ethanol concentrations (30%, 50%, 70%, 90% and 100%) for 10 min each. Subsequently, the samples were dried at the critical-point using liquid carbon dioxide, and sputter coated with gold (approximately 2 nm) using a SCD005 sputter coater (BAL-TEC, Balzers, Liechtenstein). Finally, the relevant surfaces were analyzed with a scanning electron microscope (SEM) LEO-1450 (Carl Zeiss NTS GmbH, Oberkochen, Germany), providing a rotating sample stage to allow all-around imaging.

### Odors

Odors were obtained from Sigma-Aldrich, FLUKA, Aldrich at the highest purity available. Compounds used are listed in S2 and S3 Tables, together with the respective solvent used (paraffin oil, CAS: 8012-95-1, Supelco, USA; distilled water; or ethanol, Sigma Aldrich, Germany),

in which each odor was diluted. For electroantennogram (EAG) recordings, a dilution of 10% in paraffin oil (Supelco, USA) was used, while we applied all odors at a dilution of 1% in single sensillum recordings (SSRs). In EAG recordings, certified grade carbon dioxide, diluted to a final concentration of 30% with air, was used. An odor blend, used only in SSR, was created by mixing all compounds listed in S3 Table in a 1:1 ratio, all at 1% dilution in paraffin oil. The compounds in this blend are known to be detected by odorant receptors (ORs) in other insect species, and were thus designed to identify possible ORs housed in the grooved peg sensilla of *Rhodnius spp* [55–58].

## Odor application

Odors used as stimuli were prepared at the beginning of each experimental session: a 10 μl aliquot of the diluted odor (see S2 and S3 Tables) was pipetted onto a fresh filter paper (Ø = 1 cm$^2$, Whatman, Dassel, Germany), which was placed inside a glass Pasteur pipette. Each loaded filter paper was used for a maximum of 3 times to ensure a stable concentration across experiments. Highly volatile carboxylic acids and aldehydes were loaded at each stimulus presentation. Carbon dioxide was diluted shortly before application, and a custom made syringe, which was connected to an automated stimulus controller, was used for its delivery as described before [59]. A stimulus controller (Stimulus Controller CS-550.5, Syntech, Germany) was used to deliver odors to the insect antenna through a metal pipette placed less than 1.5 cm (EAG) or 0.5 cm (SSR) away from the insect antenna. A constant humidified air flow of 1.0 l min$^{-1}$ was delivered to the insect, while each odor pulse had an airflow of 0.5 l min$^{-1}$, and was buffered with compensatory airflow of the same magnitude.

## Electroantennogram (EAG) recordings

An antenna was severed quickly between the scape and the pedicel and placed between two metal electrodes. Conductive gel (Spectra 360, Parker Laboratories, Fairfield, USA) was applied to each end of the antenna. The electrode was connected to a Syntech IDAC analog/digital converter (Syntech). Acquisition was done with Autospike32 at a sample rate of 2400 Hz. While the application of odors was randomized we did ensure to apply the control (paraffin oil) at regular intervals. During the screening of the odor panel, we observed an increase in the response amplitude to the control in function of time. To account for this bias, we normalized each recording with the following formula, similarly to how it has been previously done [60]:

$$A_n(t) = Z_n - C(t),$$

with,

$$C(t) = a\left(\frac{T-t}{T}\right) + b\left(1 - \frac{T-t}{T}\right),$$

where $A_n(t)$ is the normalized response to a given odor stimulus $n$ at a given time $t$; $Z_n$ is the measured response to the odor stimulus $n$, and $C(t)$ is the averaged solvent response at a given time $t$, with $a$ being the closest solvent response before stimulus presentation at $t_a$, and $b$ the closest solvent response after stimulus presentation at $t_b$. The contribution of each of these solvent responses to the averaged solvent response is pondered by the factor $\left(\frac{T-t}{T}\right)$, where $T = t_b - t_a$.

## Single-sensillum recordings (SSR)

Insects were placed inside a severed 5 ml plastic tip (Eppendorf, Hamburg, Germany), which was sealed with dental wax (Erkodent, Pfalzgrafenweiler, Germany). The tip was then

immobilized on a microscopy slide with dental wax. Both antennae were glued to a coverslip with double-sided tape. A tungsten electrode inserted into the insect's abdomen was used as reference. Preliminary recordings with a silver wire as a reference electrode did not show an improvement in the signal-to-noise ratio. The preparation was placed under an upright microscope (BX51WI, Olympus, Hamburg, Germany) equipped with a 50x air objective (LMPlanFI 50x/0.5, Olympus). Neural activity in the form of spike trains originating from OSNs was recorded with a sharpened tungsten electrode targeted at the base of a grooved peg sensillum. Signals were amplified (Syntech Universal AC/DC Probe; Syntech) and sampled at 10,666.7 samples s$^{-1}$ through an USB-IDAC (Syntech) connected to a computer. Spikes were extracted using Autospike32 software (Syntech). Odor responses from each sensillum were calculated as the difference in the number of impulses 0.5 s before and after stimulus onset.

The response to each odorant in the SSR recordings was calculated as the change in spikes s$^{-1}$ upon odor stimulation using Autospike32. The response to the solvent was subtracted from each measurement. The number of OSNs housed in each sensillum in *R. prolixus* has been estimated to be between 5 and 6 [29, 61]. We attempted to confirm this observation using semi-thin section of the antenna, but, despite our efforts, were unable to decisively identify the number of sensory neurons in the grooved peg (GP) sensilla in any of the two species. For that reason, we decided to define each sensillum as a responding unit, as it has been done in other insects [62].

Subsequent analysis was carried out in MATLAB (The MathWorks Inc, Natick, USA) in which an agglomerative hierarchical clustering of the sensillum responses, with a Euclidean metric and Ward's method, was performed. The inconsistency coefficient was calculated for each link in the dendrogram, as a way to determine naturally occurring clusters in the data [63, 64]. A depth of 4 and a coefficient cutoff of 1.8 for *R. prolixus* and 1.0 for *R. brethesi* were used in the calculation.

The response of each sensillum type was taken as the average response of individual sensilla belonging to the same cluster (i.e., sensillum type). The average responses of each sensillum type were then grouped and averaged for the chemical classes. These responses were then normalized to the maximum response within each sensillum type. Principal component analysis (PCA, with a Singular Value Decomposition (SVD) algorithm) was performed in MATLAB (The MathWorks Inc) using the averaged scaled (between 0 and 1) and z-score normalized responses for each sensillum type and each species. As a measure for similarity, we applied a One-Way ANOSIM to calculate whether different sensillum types represent significantly different classes [65].

Averaged responses were computed as the mean of all sensillum responses to a particular odorant. To compare among chemical classes, these odor responses were then further averaged for each particular chemical class. Comparison between species was done using unpaired two-tailed Student t-tests (GraphPad Prism 8, San Diego, USA). These average responses were then normalized to the odor that elicited the hightest responses in each species, being for both species propionic acid, and the lifetime sparseness (S) of each sensillum was calculated. We applied the lifetime sparseness as a measure of the response breadth of each sensillum. For its calculation the following formula was used [66]:

$$S = \left(\frac{1}{1-\frac{1}{N}}\right) * \left(1 - \frac{\left(\sum_{j=1}^{N} r_j/n\right)^2}{\sum_{j=1}^{N} r_j^2/N}\right),$$

where S is the lifetime sparseness, *N* is the number of tested odors and $r_j$ is the sensillum response to any given odor *j*, with $r_j \geq 0$ and $S \in [0,1]$, where *S = 0* corresponds to the case in

which the sensillum responds equally to all odorants, and $S = 1$ to where the sensillum responds to only one odor of the set.

## Results

### Species-specific morphological differences of the antenna

Previous studies have demonstrated that sensillum patterns in haematophagous insects, including Triatominae species, reflect specific adaptations to different hosts and habitats [67–70]. Through a comparative qualitiative and quantitative analysis of the main olfactory organ, the antenna, we assessed potential morphological differences between the generalist *R. prolixus* and the sylvatic specialist *R. brethesi*, using scanning electronic microscopy (SEM) (Figs 1, S1 and S2). A qualitative analysis of the morphological patterns of sensilla on the antenna of both species did not reveal major differences. For further analysis, sensilla were classified according to the study by Shanbhag et al. [71], as it has been previously done in triatomines [30, 33]. In both species, the second segment, or pedicel, was found to be enriched with sensilla described to have a mechanosensory function [38–40]: sensilla trichobothrium, bristles I and III (S1 and S2 Figs). Moreover, the cave organ, a sensillum type shown to have a thermo-receptive function in *R. prolixus* and *T. infestans* [72, 73] was also found in *R. brethesi* (S2B Fig). However, we were unable to identify the previously described ornamented pore [38] in *R. brethesi*, possibly due to the angle of orientation of our preparation. Notably, in one preparation, our micrographs show the existence of two previously undescribed sensillum types on the pedicel of *R. prolixus*. One is a peg-in-pit sensillum with an inflexible socket, with no evident pores, housed within a chamber in the antennal cuticle (S1B Fig). This type is reminiscent of the thermosensitive sensilla coeloconica [42], but its function remains unknown. The second sensillum type resembles a type 3 coeloconic sensilla, characterized by two pores at its base, described in other hemipteran species (S1E Fig) [45].

Wall-pore sensilla with inflexible sockets were found on flagellomere I and the distal half of flagellomere II in both species. These include the trichoid and basiconic sensilla (also known as thick- and thin-walled trichoid sensilla [74], respectively), as well as the double-walled grooved peg sensilla (referred to as basiconic sensilla in [74]) (Fig 1C and 1D). All of these sensillum types show slight longitudinal grooves filled with pores or slits indicative of an olfactory function [30]. On the same antennal segments of both species we also identified a poreless sensillum with an inflexible socket: the campaniform sensillum [74, 75] (Fig 1C and 1D).

As shown in other studies, quantitative differences in the number of olfactory sensilla of the species potentially reflect particular adaptations to their ecological niche. Thus, we performed autofluorescence confocal scans of glycerol-embedded antennae in order to quantify the sensillum density on the flagellomere II, where putative olfactory sensilla reach the highest density, and inter-specific differences were previously reported among triatomines [45] (Fig 2). Our results show that the density of both basiconic and trichoid sensilla is significantly higher in *R. brethesi* than in *R. prolixus*. In contrast, the density of grooved peg sensilla was not significantly different between the two species.

### Antennal responses in *Rhodnius prolixus*

Previous studies aimed at characterizing odor-evoked responses in triatomines have focused on *T. infestans*, and reported the response to a small number of chemical compounds, comprising aldehydes, acids and amines [30, 32, 36]. To assess whether other chemical classes are detected by the antennae of of triatomines, we performed EAG recordings in *R. prolixus* using a panel of 27 odors, belonging to various chemical classes, that have previously been shown to elicit behavioral responses in triatomines and other haematophagous insects (Fig 3 and S1 and

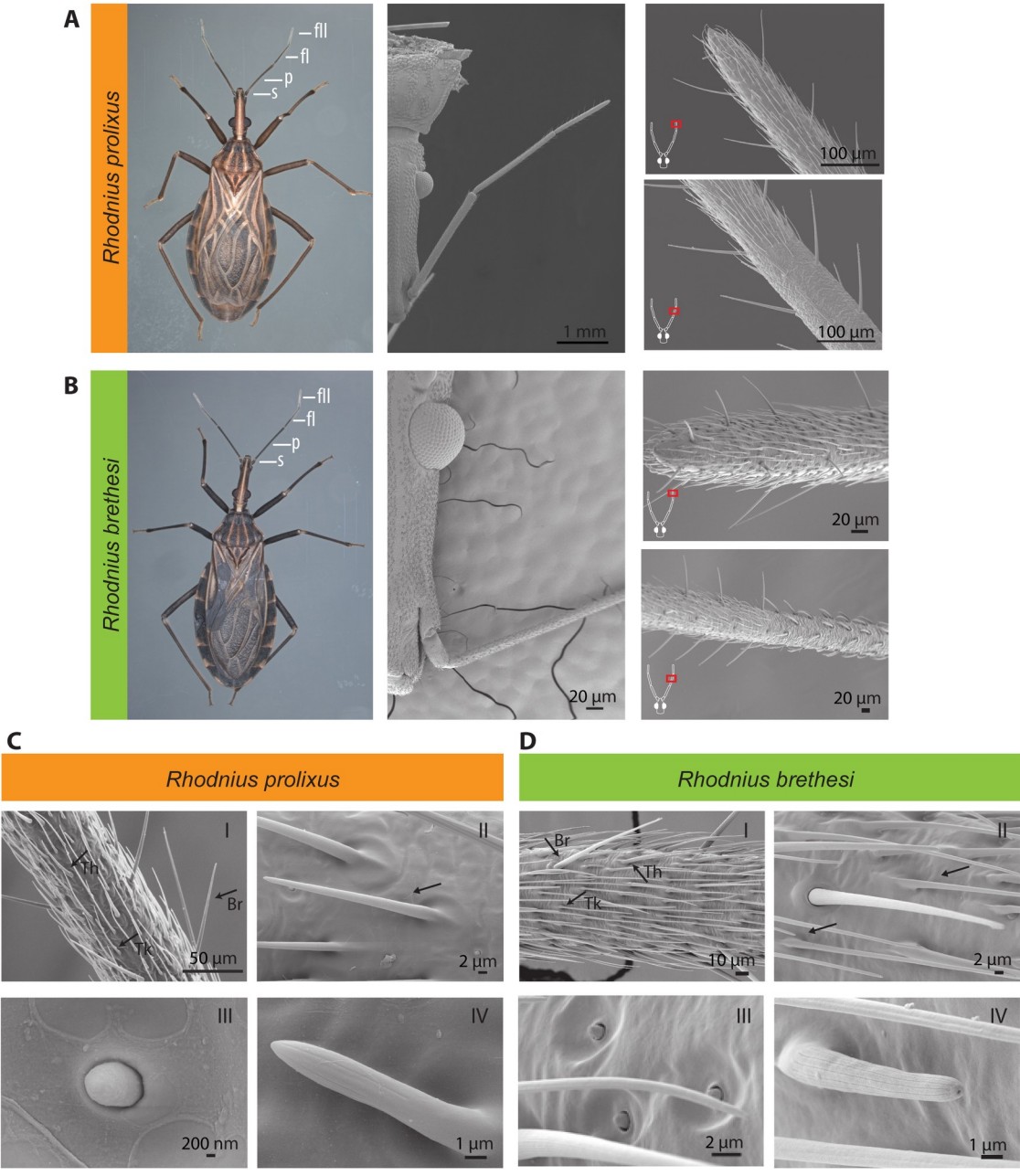

**Fig 1. Morphology of the peripheral olfactory system of *Rhodnius prolixus* and *Rhodnius brethesi*.** (A,B) Light-microscopic images of whole body of *R. prolixus* and *R. brethesi* (left panel). Antennal segments indicated: scape (s), pedicel (p), flagellomere I (fI), and flagellomere II (fII). Scanning electron microscopy (SEM) images of head (middle panel) and corresponding antennal segments (right panel). (C,D) Antennal sensilla of *R. prolixus* and *R. brethesi* on flagellomere II: arrows indicate (I) thick-walled (Tk), and thin-walled (Th) trichoid sensilla (also known as trichoid and basiconic sensilla), and bristles (Br). (II) Thick-walled trichoid sensillum, (III) campaniform sensillum, and (IV) grooved peg sensillum.

S2 Tables). Significant responses (p<0.05) were observed for 33% of the tested odors (one sample t test against zero). Out of these, the strongest response was observed to acetic acid, a compound that is present in triatomine feaces and mediates aggregation [76], followed by propionic acid, a known host volatile, to which *T. infestans* is behaviorally active [33]. Significant responses were also seen to the main component of the alarm pheromone [77], isobutyric acid,

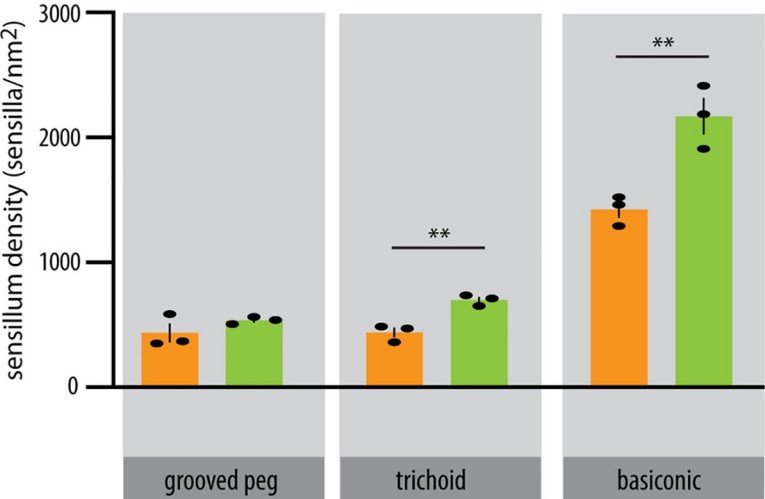

**Fig 2. Sensillum density in *Rhodnius prolixus and Rhodnius brethesi*.** Density of olfactory sensilla in flagellomere II for each species estimated as the total number of sensilla for each type, by the flagellar surface area, n = 3, unpaired t test: \*\*p<0.01. Data represents mean ± SEM.

a compound that is also present in host volatiles [27], and to the closely related compound butyric acid. Taken together, the responses to acids represented 44% of the total significant responses. Additionally, *R. prolixus* showed a significant, though smaller, response to other host volatiles, such as cyclohexanone, amyl acetate and trimethyl amine.

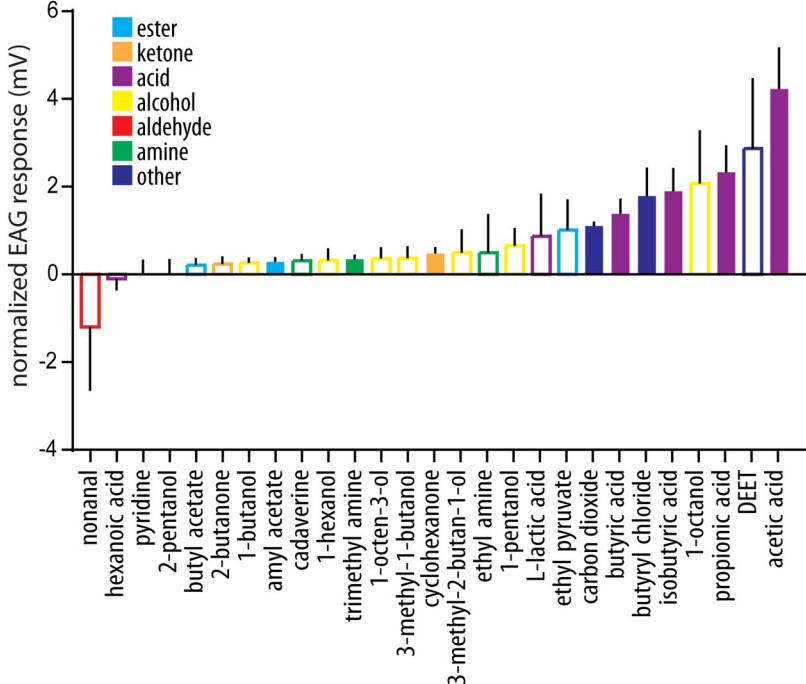

**Fig 3. Antennal responses from *Rhodnius prolixus*.** Ecologically relevant odorants (see S1 Table) were applied to the insect antenna, and evoked responses were measured through electroantennogram analysis. Responses were normalized to their corresponding solvent response (see Methods). Filled bars represent statistically significant responses (one-sample t test against zero: p<0.05, n = 7–10 for each odor tested).

A significant response was recorded for 30% carbon dioxide, a chemosensory cue that is attractive for *T. infestans* at lower concentrations [20]. Interestingly, we also observed a significant olfactory response to butyryl chloride. While this compound is proposed to act as an insect repellant, as it inhibits the activity of the carbon dioxide-detecting sensory neurons in mosquitoes [60], its function and detection in triatomines has not been studied so far.

## Odor responses in grooved peg sensilla

The EAG recordings of *R. prolixus* demonstrated that the olfactory system of these insects responds mostly to acids and amines. These compounds are commonly found in the environment of the insects (S1 Table), and their role in regulating odor-guided behavior has been assessed for some species of triatomines [19, 21, 22, 54]. In insects, acids and amines are detected by neurons housed in antennal grooved peg (GP) sensilla [78–80]. As shown in our morphological studies, this sensillum type is present in the antenna of both *Rhodnius* species at a low density, making it an ideal system to assess species-specific differences in olfactory tuning.

To assess the tuning of individual GP sensilla, here defined as a responding unit, we tested a total of 38 odors, out of which 17 were acids and 9 amines, varying in carbon length and branching. We included additional volatiles (such as indole and amyl acetate) known to be present in, but not exclusive to, vertebrate hosts, or previously shown to be detected by GP sensilla in other insects [25, 36]. In addition, a custom OR blend (S3 Table), composed of compounds typically detected by odorant receptors (ORs) in other species [55–58] was also applied. Averaged sensillum responses demonstrated that *R. brethesi* responded generally stronger to odors than *R. prolixus* (Fig 4A). In addition, a significant overall interspecific difference was found for 58% of the odorants. While both species exhibited the strongest response to propionic acid, major differences were seen for the following compounds: butyric acid, benzaldehyde, valeric acid, 2-oxopropanoic acid, formic acid, and for the OR blend, with *R. brethesi* displaying a higher response than *R. prolixus* in all cases. Butyryl chloride was the only compound with a significantly higher response in *R. prolixus*. Stimulation with palmitic acid generated the strongest inhibitory response in both species. When responses were normalized to the maximum odor response (i.e. propionic acid in both species), significant differences remained for five odorants: butyraldehyde, butyryl chloride, amyl acetate, 3-methyl indole, and for the OR blend.

*Rhodnius prolixus* responded, on average, more frequently to amines than *R. brethesi* accounting for 40% of responses in *R. prolixus* compared to 24% in *R. brethesi* (Fig 4B). In contrast, *R. brethesi*, responded more strongly to aldehydes, with 33% compared to 16% in *R. prolixus*. The responses to acids within each species were comparable and accounted for 18% in *R. prolixus* and 22% in *R. brethesi*. Similar results were found for the mixed chemical category (*i. e.*, 'other') with 18% in *R. prolixus* and 15% in *R. brethesi*. Finally, averaged responses of *R. brethesi* to esters were slightly higher than those found for *R. prolixus* (14% to 8%).

## Response tuning of grooved peg sensilla

In order to quantify and compare the tuning width of the GP sensilla between both species, we plotted the species-specific tuning curves and determined the lifetime sparseness (S) (Fig 4C). The lifetime sparseness is usually calculated to assess how broadly or narrowly tuned olfactory receptors are, while in our case this serves as a measure of the GP-sensillum tuning. This analysis demonstrates that *R. prolixus* is indeed tuned to a narrower selection of odors than *R. brethesi* with an S-value of 0.5 for *R. prolixus* compared to 0.35 in *R. brethesi*.

We next wondered whether the stronger responses observed for *R. brethesi* result from a higher proportion of individual sensilla showing excitatory odor-evoked responses or, less

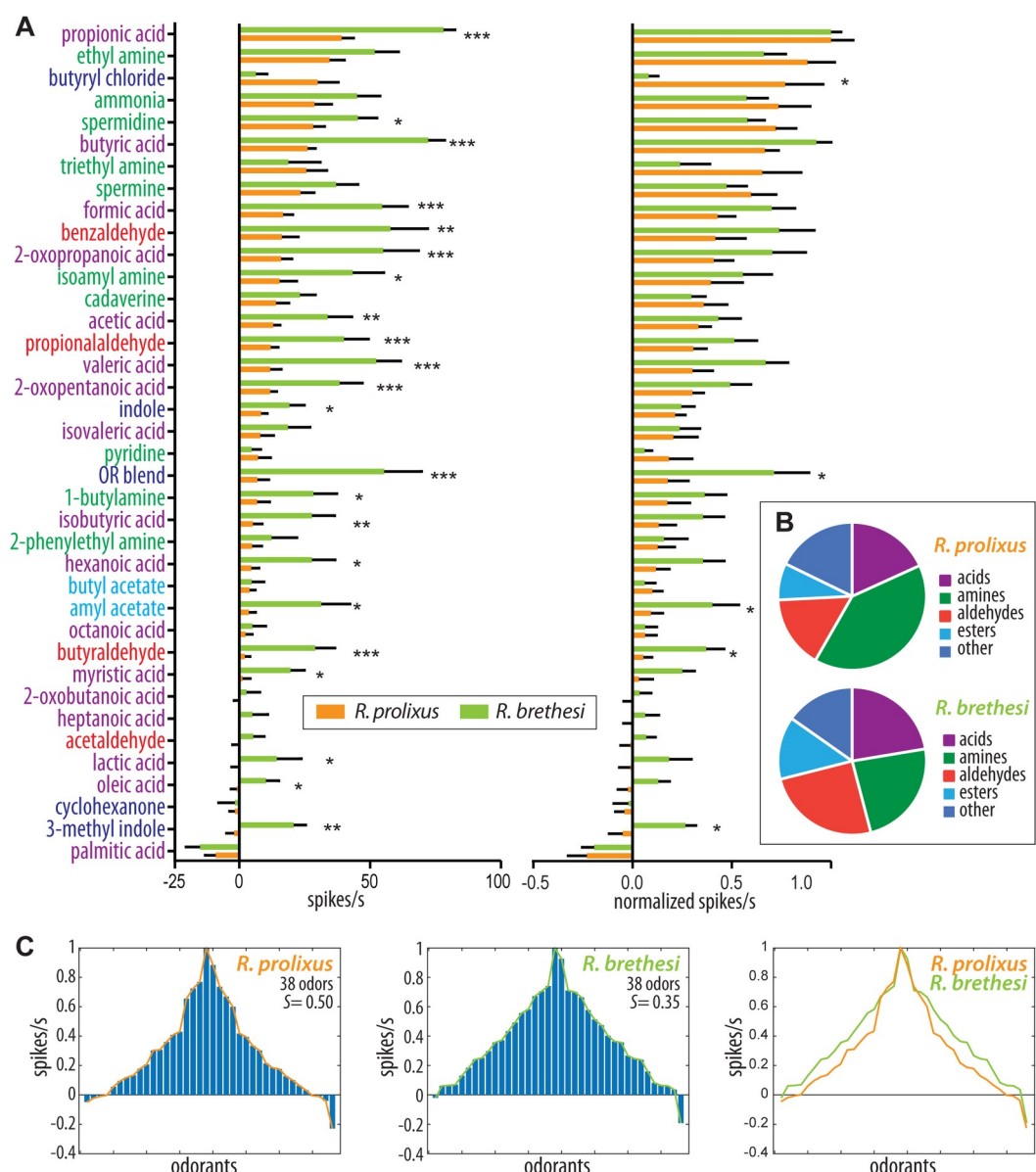

**Fig 4. Grooved peg sensillum responses to ecologically relevant volatiles in *Rhodnius prolixus* and *Rhodnius brethesi*.**
(A) Averaged odor responses from the grooved peg sensilla from *R. prolixus* and *R. brethesi*. Responses were normalized to the maximum response recorded, which was propionic acid for both species. (B) Pie-charts represent averaged responses according to chemical class, for both species. (C) Tuning curves for each species. Odors that elicit the weakest responses are placed at the edges. The order of the odors is different for the different species. The lifetime sparseness (*S*) was calculated for each species (see Methods).

likely, a decrease in inhibitory sensillum responses in *R. brethesi* compared to *R. prolixus*. Thus, in order to further characterize these responses, we analyzed single odor-sensillum combinations. Since each GP sensillum was screened with a comprehensive odor panel composed of a total of 38 odors, our SSR data comprised 950 odor-sensillum combinations in *R. prolixus*, and 380 odor-sensillum combinations in *R. brethesi*. While in *R. prolixus*, only 31% of these odor-sensillum combinations yielded responses >15 spikes s⁻¹ above solvent response, 60% did in *R. brethesi* (Fig 5). This difference was also consistent at responses with higher spike

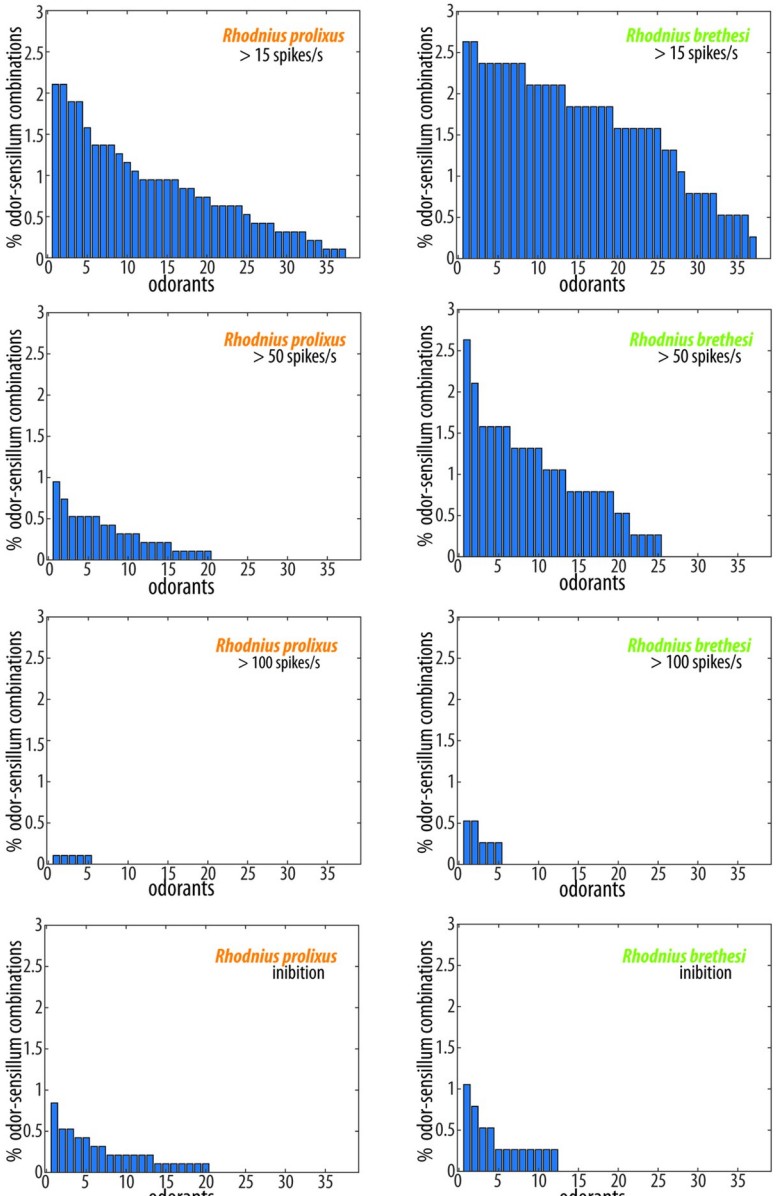

**Fig 5. Response properties of grooved peg sensilla.** Percentage of odor-sensillum combinations activated at the indicated firing rate by each odorant. The total number of odor-sensillum combinations was 950 for *R. prolixus* and 380 for *R. brethesi*. Odorants are sorted along the x-axis according to the number of sensilla that they activate. Responses with a frequency under -15 spikes/s are considered inhibitory.

frequencies ($>50$ spikes s$^{-1}$): in *R. prolixus*, only 7% of these combinations held responses higher than 50 spikes s$^{-1}$ above solvent, while in *R. brethesi* 26% of the odor-sensillum combinations resulted in responses of $>50$ spikes s$^{-1}$. Responses above 100 spikes s$^{-1}$ were generally scarce in both species.

Inhibitory responses were less prevalent than excitatory ones, with only 5% of the odor-sensillum combinations identified as inhibitory ($<$-15 spikes s$^{-1}$ compared to solvent control), in both *R. prolixus* and *R. brethesi*. Inhibition could not be attributed to a single odorant since 53% of the odors in the panel generated at least one odor-sensillum inhibition in *R. prolixus*,

and 32% of the odors resulted in an inhibition in *R. brethesi*. Taken together, our data suggests that the stronger responses seen *in R. brethesi* can be attributed to a higher proportion of responses being above 15 spikes s$^{-1}$, and not to a difference in inhibitory responses between these species.

### Funcional classification of grooved peg sensilla

To further assign the measured odor responses to distinct and functional GP sensillum sub-types in each of the two species, we performed an agglomerative hierarchical clustering analysis (Fig 6). Responses could be clustered into 4 groups in each species, corresponding to putative functional sensillum types classified as GP1 to GP4. It should be noted that, in both species, all of the sensillum types responded to butyric acid, as well as to propionic acid. In particular, strong responses (i.e. > 50 spikes s$^{-1}$) to acids were more prominent in *R. brethesi*, with all of the sensillum types responding to at least 7 out of the 17 acid compounds tested. A major difference between the species was the response to our custom OR blend. While only one sensillum responded to it in *R. prolixus* (with >50 spikes s$^{-1}$), 50% of the sensilla showed a response to the blend in *R. brethesi*.

   As each of the four putative sensillum types responded to a particular combination of odors (Fig 6), we propose these as diagnostic odors for each specific GP type (Fig 7A). In *R. prolixus*, GP type 1 (Rp-GP1), which accounts for 40% of the GP-sensilla recorded from, responds preferentially to the amines trimethylamine, ammonia and ethylamine, as evidenced by the average responses. Rp-GP2 comprises 16% of the GP-sensilla and responds best to propionic acid, triethylamine, spermine, spermidine and benzaldehyde. Rp-GP3 shows the highest responses to isoamylamine and butyryl chloride and stands for 28% of the GP-sensilla, while Rp-GP4, representing 16% of the sensilla, responds to ammonia, ethylamine and butyryl chloride.

   In *R. brethesi*, the type 1 GP-sensillum (Rb-GP1) responded preferentially to butyric acid and was inhibited by amyl acetate (Fig 7A). The Rb-GP2, with a similar response profile to Rb-GP1, differed from it in the responses to amyl acetate and 2-oxopropanoic acid. It also showed a higher response to isoamyl acetate, butyric, valeric and formic acids than Rb-GP1. The Rb-GP3 type showed high responses to 2-oxopropanoic acid and formic acid. Finally, type GP4 of *R. brethesi* showed a strong response to benzaldehyde, ammonia and propionaldehyde. Rb-GP1 represented 30%, Rb-GP2 20%, Rb-GP3 20% and Rb-GP4 30% of the total number of grooved peg sensilla recorded from in this species.

### Odor tuning to chemical classes

Next, we analyzed whether the individual sensillum types respond preferentially to particular chemical classes (Fig 7B). In *R. prolixus*, Rp-GP1 responded strongest to amines, Rp-GP2 to aldehydes and to a lesser extent to amines. Rp-GP3 did not respond preferentially to any chemical odor class, with most responses being to butyryl chloride, and Rp-GP4 showed the strongest responses to amines. In *R. brethesi*, all of the sensillum types responded to, at least, two of the chemical classes tested. While both Rb-GP1 and Rb-GP3 showed the strongest responses to acids, Rb-GP3 but not Rb-GP1 responded additionally to aldehydes. Rb-GP2 did not respond to any particular odor class, with its highest responses shown to the OR blend. Finally, Rb-GP4 responded mainly to aldehydes, followed by amines.

   We next evaluated whether odor compounds with a certain carbon length evoked stronger responses in the *Rhodnius* grooved peg sensilla by focusing on C1-to-C18 of acids and amines (Fig 7C). In *R. prolixus*, we observed higher responses for short chain carboxylic acids (C1-6/7), with three of the sensillum types showing a negative correlation between carbon chain length and response strength (Pearson correlation; Rp-GP1: r = -0.87, p = 0.0005; Rp-GP2: r =

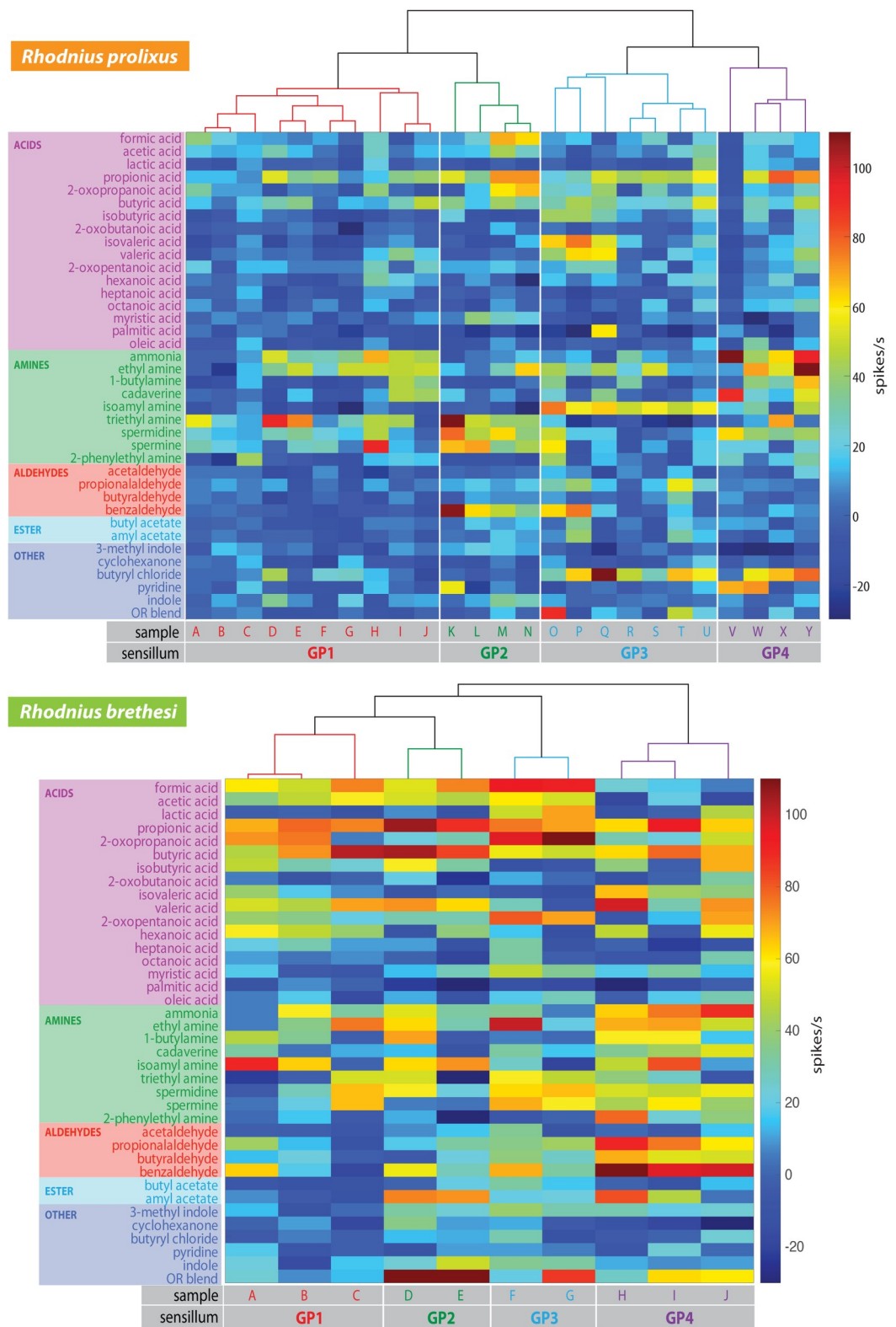

**Fig 6. Functional classification of grooved peg sensilla in *Rhodnius prolixus* and *Rhodnius brethesi*.** Color-coded responses from the grooved peg sensillum (flagellomere II) in *R. prolixus* (n = 25) and *R. brethesi* (n = 10) to the odor panel. In each case, the dendrogram represents the agglomerative hierarchical clustering (Ward's method, Euclidean distance) of these responses.

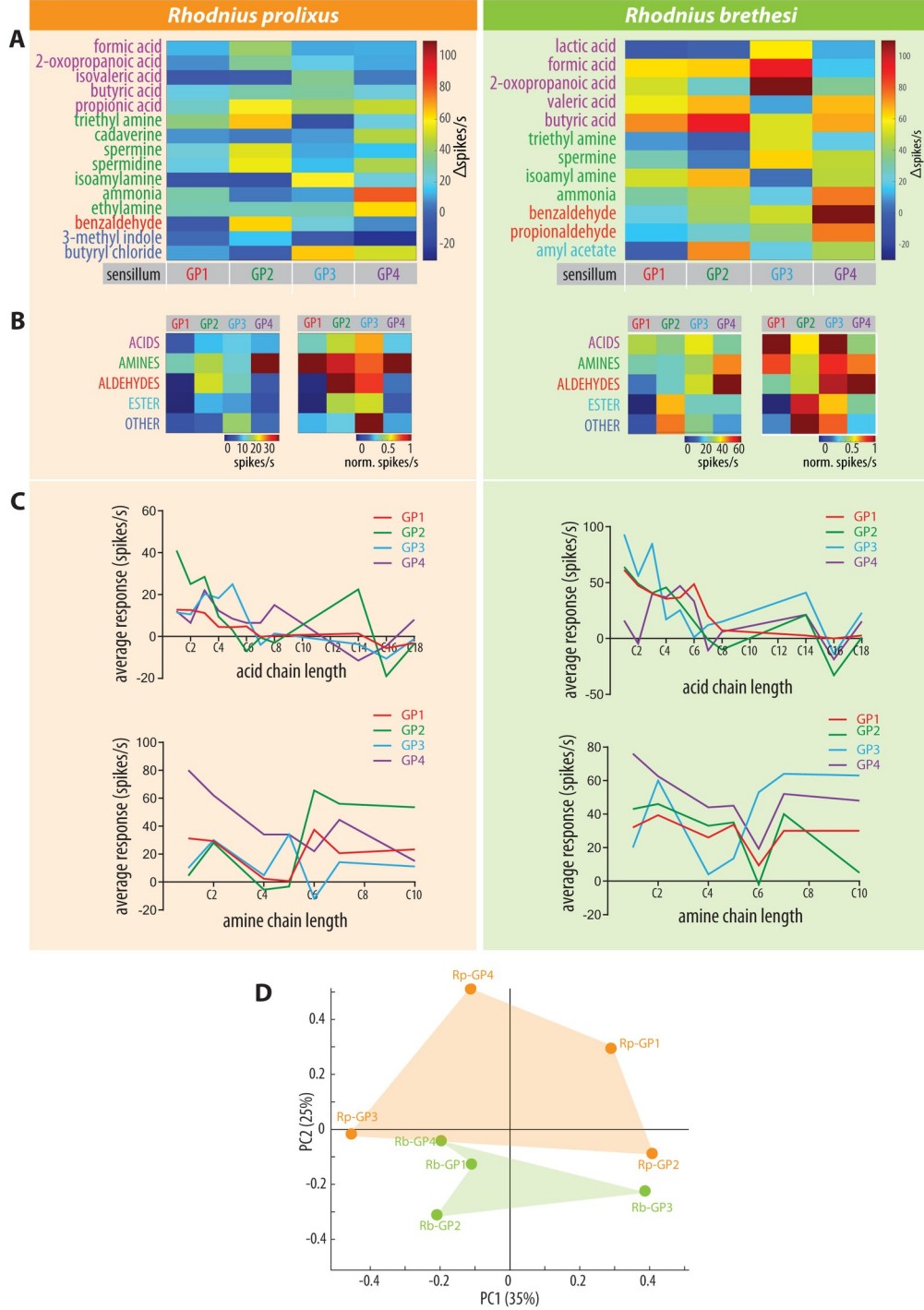

**Fig 7. Response profile of the functional subtypes of grooved peg sensilla.** (A) Color coded mean responses for the selected diagnostic odors for each sensillum type, for each insect species. Odors were chosen as diagnostic when they allowed the maximum separation between sensillum types. (B) Averaged responses for each chemical class and sensillum type (left panel), normalized to the maxiumum response within each sensillum type (right panel). (C) Averaged responses (n = 2–6) to acids and amines with the indicated carbon chain length. In *R. prolixus*, GP1, GP3, GP4 show a negative correlation between carbon length and response strength to acids (Pearson correlation; Rp-GP1: r = -0.87, p = 0.0005; Rp-GP2: r = -0.59, p = 0.054; Rp-GP3: r = -0.75, p = 0.008; Rp-GP4: -0.61, p = 0.045, n = 11). In *R. brethesi* GP1 and GP2 show a significant negative correlation between carbon length and response strength to acids (Pearson correlation; Rb-GP1: r = -0.89, p = 0.0002; Rb-GP2: r = -0.76, p = 0.006, n = 11). (D) Principal component analysis (PCA) of *R. prolixus* and *R. brethesi* sensillum types. No significant difference was observed for the sensillum space of the two species (ANOSIM, p = 0.32, Euclidean).

-0.59, p = 0.054; Rp-GP3: r = -0.75, p = 0.008; Rp-GP4: -0.61, p = 0.045, n = 11). Interestingly, GP2 of *R. prolixus* showed weaker responses to short chain amines, but stronger ones to those with long chains (C6-C10). Acid carbon chain length also appeared to be relevant for *R. brethesi*, where it was negatively correlated with response intensity in 2 out of the 4 sensillum types (Pearson correlation; Rb-GP1: r = -0.89, p = 0.0002; Rb-GP2: r = -0.76, p = 0.006, n = 11). In contrast, in the case of the amines, a decrease in activity with increasing carbon length was seen in GP4 (Pearson correlation; r = -0.85; p = 0.016, n = 7) in *R. prolixus* but not in other GP sensillum types of *R. brethesi*. However, when compared to *R. prolixus*, *R. brethesi* displayed stronger responses to short chain (C1-C5) amines (*R. prolixus*: 19.63 ± 2.65, n = 125; *R. brethesi*: 19.63 ± 2.65, n = 51; unpaired t test, p = 0.0005).

Finally, we addressed the comparability of the described functional sensillum types between species. In order to get a notion of similarity between the GP types described, we calculated the Euclidean distances between the sensillum types for the two species (S4 Table). The averaged response values were first z-score normalized (mean = 0, standard deviation = 1), to ensure that the distance measured reflects dissimilarities between response patterns and not magnitude. The sensillum pair that showed the lowest distance was GP2 in *R.prolixus* (Rp-GP2) and GP3 in *R. brethesi* (Rb-GP3; distance = 4.47). The pair Rp-GP4 and Rb-GP3 was on the other end of the spectrum, with the highest distance (8.24). In between we found most (88%) of the sensillum combinations to be within the range of 6–8.3 units of distance. In order to further explore the differences between the two species, we performed a principal component analysis (PCA) in which the 38-dimensional sensillum space was reduced to lower dimensions (Fig 7D). We focused on the first two components, which together explain 60% of the variance. While the sensillum types of *R. brethesi* appeared to be more densely clustered than the ones of *R. prolixus*, the distance between individual sensillum types was larger within an individual species than between species (ANOSIM, R = 0.09, p = 0.32). Taken together, these results show that sensillum subtypes are not necessarily species-specific, despite each showing a different odor tuning breath and responding to a specific set of ligands.

## Discussion

In this study morphological and functional differences in the peripheral olfactory system of *R. prolixus*, and *R. brethesi*, two species differing in distribution and refuge habitat, were assessed. Morphological differences, in terms of a higher density of basiconic and trichoid sensilla in *R. brethesi* compared to *R. prolixus*, were found, a character previously described in other triatomines [45, 46, 48, 69, 81]. A correlation between the number of olfactory sensilla and habitat range has been proposed for triatomines, as well as for other haematophagous insect species [68]. However, this remains to be confirmed for *R. prolixus* and *R. brethesi*, as rearing conditions may negatively affect sensillum numbers [48, 49]. However, our results are intriguing as *R. brethesi* is a refugee specialist, nesting in only one species of palm tree, in sylvatic environments, suggesting that this species may have a different need or uses a different strategy for detecting and discriminating odors, compared to *R. prolixus*.

To assess the olfactory function of the *R. prolixus* antenna we initially used EAG analysis with known biologically active compounds, previously shown to be involved in intraspecific communication and other odor-guided behaviors. Surprisingly, the antenna of *R. prolixus* responded only to a limited number of the compounds tested. For instance, we did not see a significant response to either 2-butanone or 3-methyl-2-butanol, compounds known to be part of the sexual pheromone [82]. We hypothesize that the lack of antennal response is likely a consequence of the low number of specialized neurons detecting these compounds, and that the limited sensitivity of the EAG analysis failed to provide a reliable signal. However, it might

be possible that these chemicals are detected by organs other than the antenna, as several odorant and iontropic receptors (ORs and IRs), along with the odorant co-receptor orco are expressed also in tarsi, genitalia and rosti [83]. Most of the odorants evoking a significant antennal response were volatiles characteristic of the vertebrate (amniote) odor signature, such as acetic acid, propionic acid, butyric acid, isobutyric acid, ethyl pyruvate, trimethyl amine and carbon dioxide. All of these volatile compounds have been identified in the headspace of vertebrates, and males and females of *R. prolixus* have been demonstrated to be attracted to acetic and isobutyric acid [21]. These compounds, however, in addition to often occurring in vertebrate host secretions, are also used in intraspecific communication [84], highlighting the importance of sensory parsimony in these insects [17].

In insects, ORs and IRs are responsible for the detection of volatile molecules. IRs are thought to be ancestral, as they are found in basal insects and in their most recent phylogenetic antecessor [85, 86]. These receptors, expressed in the dendrites of OSNs housed in grooved peg sensilla (i.e. double walled sensilla), serve a conserved function in the detection of acids and amines across insect taxa [33, 36]. Yet, we show that triatomine insects, with different habitat and host requirements, show differences in the olfactory tuning of their GP sensilla. While both species respond to acids and amines varying in branch and carbon length, *R. prolixus* appears to be more tuned to amines than its sylvatic sibling. This result goes in line with a previous study showing that *R. prolixus* is attracted to amines present in vertebrate-hosts excresions which guide their host search [34].

Both species differed in their responses to certain odorants. *R. prolixus* displayed a significantly stronger response to butyryl chloride than *R. brethesi*. It is assumed that this odor compound has a repellant function in mosquitoes by inhibiting the activity of carbon dioxide-responding neurons [60]. Whether butyryl chloride serves a similar role in triatomines requires confirmation. Interestingly, *R. brethesi* revealed higher responses to amyl acetate, a compound found in fruits [87, 88], 3-methyl indole, which occurs in feces and in inflorescences at low concentrations [89–91], butyraldehyde and to the OR blend. Previous studies have shown that compounds present in this blend are detected by ORs in other insect species [55–58]. While only one of the sensilla probed in *R. prolixus* responded to this blend, half of them did in *R. brethesi*. This suggests that ORs may be present in the grooved peg sensilla of *R. brethesi* but not of *R. prolixus*, as is the case, for instance, for the odorant receptor OR35a, expressed in the coeloconic sensillum of *Drosophila melanogaster* [79]. Overall, these differences might reflect specific adaptations to their corresponding environments.

Based on their odor response profiles, we identified four functional sensillum subtypes in each species. This is in contrast with studies on *T. infestans*, where only three grooved peg sensillum types were described for 5th instar nymphs [92]. This discrepancy might be due to several reasons. First, it is possible that differences among triatomine species are larger than expected, as suggested e.g., by ultra-structural studies demonstrating different number of OSNs in GP sensilla of triatomines [38, 61]. Second, different patterns of behavior in response to odorants are also recognizable between *T. infestans* and *R. prolixus* [32]. Third, we recorded responses of adults, whereas 5th instar nymphs were examined in the case of *T. infestans*. The antenna of *R. prolixus* undergoes significant changes between the 5th instar and adult, with an increased number of olfactory sensilla on flagellomeres I and II [74], probably related to intraspecific communication or behavioral needs. These changes might account for the additional sensillum subtype observed in adults of *Rhodnius*. Fourth, and lastly, in our screening we recorded responses from a higher number of chemicals than in previous studies [36], potentially improving the resolution of physiological sensillum subtypes. It should be noted here, however, that our investigation is not conclusive, and recordings with additional compounds might help to complete the ongoing work of sensilla classification in *R. prolixus* and *R. brethesi*.

Furthermore, it would be interesting to analyze dose-response characteristics to identify the best odor ligands for the different GP sensilla types in future studies.

Interestingly, *R. brethesi* presented overall higher and broader olfactory responses than *R. prolixus*, as reflected in the average response, sensillum odor tuning and lifetime sparseness in SSR data, suggesting a sensory differentiation between these species. In insects, the number of olfactory receptors and the complexity of the ecological niche seems to be highly correlated, with the number of ORs increasing with niche complexity. For example, while tsetse flies have only 40–46 ORs, eusocial insects like ants possess over 350 ORs [93–95]. Moreover, in mosquitoes, host preference has been suggested to account for differences in the chemosensory gene repertoire between sibling species [96]. Notably, in triatomines, olfactory binding proteins (OBPs) and chemosensory proteins (CSPs) are present at lower expression levels in domestic insects of *T. brasiliensis*, compared to sylvatic and peridomestic ones [50]. Given that *R. prolixus* is not exclusively domiciliated, and the current lack of data on the chemical cues that sylvatic triatomine species encounter in the wild, we are unable to conclusively determine whether the observed differences reflect an adaptation of the two *Rhodnius* species to their specific habitats. Future experiments, comparing wild and domiciliated individuals of *R. prolixus* are required to further shed light on this hypothesis. It is important to note that triatomines process odor information in the context of other sensory cues [30]. In fact, *R. prolixus* presents an astounding thermosensitivity, and heat represents the main host-associated cue for these insects [23, 97–99]. Therefore, in the context of our results, it might be that *R. prolixus* relies less heavily on olfactory cues than *R. brethesi*, similar to what previously has been observed in other insect species [100].

To conclude, our results confirm previous observations of phenotypic plasticity in the *Rhodnius* genus. We demonstrate that the species not only differ in the morphology of their sensory equipment, but also functionally, with *R. prolixus* presenting a distinctly decreased olfactory function. It is likely that the condition found in the sylvatic species represents the ancestral character state in the subfamily, whereas a derived reduced condition might be associated with changes in habitat preference. With the ongoing rapid destruction of natural environments [8], it is likely that more species will follow this path. Careful analyses of differences and potential shifts in the sensory apparatus may turn out as helpful in the design of efficient future vector control strategies.

## Supporting information

**S1 Fig. Scanning electron microscopy (SEM) of antennal sensilla of *Rhodnius prolixus*.**
Arrows indicate (A) sensillum trichobothrium (I) and bristle II (II), (B) peg-in-pit sensilla, (C) bristle III, (D) ornamented pore, and (E) type 3 coeloconic sensilla, on the pedicel of the antenna.
(TIF)

**S2 Fig. Scanning electron microscopy (SEM) of antennal sensilla of *Rhodnius brethesi*.**
Arrows indicate (A) sensillum trichobothrium, (B) cave organ at the pedicel (I) and bristle III (II), (B') detail of the cave organ, (C) basiconic (also known as thin-walled trichoid) sensillum presenting the ecdysis channel, (D) coeloconic sensillum.
(TIF)

**S1 Table. Description of the odor panel used in electrophysiological experiments.**
(XLSX)

**S2 Table. Chemical compounds used in EAG recordings.**
(TIF)

**S3 Table. Chemical compounds used in SSR recordings.**
(TIF)

**S4 Table. Euclidean distance for the z-score normalized sensillum types of *R. prolixus* and *R. brethesi*.**
(TIF)

## Acknowledgments

We are thankful to Veit Grabe, Katharina Schneeberg, and Sandor Nietsche for help in morphological analysis, Claudio Lazzari and Marcelo Lorenzo for helpful discussions, Günther Schaub for continuously providing insects and giving useful information on insect handle and care, and Norisa Meli for assistance in insect breeding.

## Author Contributions

**Conceptualization:** Florencia Campetella, Rickard Ignell, Bill S. Hansson, Silke Sachse.

**Data curation:** Florencia Campetella, Silke Sachse.

**Formal analysis:** Rickard Ignell, Rolf Beutel, Silke Sachse.

**Funding acquisition:** Florencia Campetella, Bill S. Hansson, Silke Sachse.

**Investigation:** Florencia Campetella.

**Methodology:** Florencia Campetella.

**Project administration:** Silke Sachse.

**Resources:** Rickard Ignell, Rolf Beutel, Bill S. Hansson, Silke Sachse.

**Software:** Florencia Campetella.

**Supervision:** Bill S. Hansson, Silke Sachse.

**Validation:** Florencia Campetella.

**Visualization:** Florencia Campetella, Silke Sachse.

**Writing – original draft:** Florencia Campetella, Silke Sachse.

**Writing – review & editing:** Florencia Campetella, Rickard Ignell, Rolf Beutel, Bill S. Hansson, Silke Sachse.

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
