## [Decision Letter · Decision Letter 0]

5 Feb 2021

Dear Dr Sachse,

Thank you very much for submitting your manuscript "Comparative dissection of the peripheral olfactory system of the Chagas disease vectors Rhodnius prolixus and Rhodnius brethesi" for consideration at PLOS Neglected Tropical Diseases. As with all papers reviewed by the journal, your manuscript was reviewed by members of the editorial board and by independent reviewers. As you can see in the reviews (below this email), the reviewers raised some substantial concerns about the manuscript as it currently stands, including the construction of the study hypothesis (raised by reviewer 2). These issues must be addressed before we would be willing to consider a revised version of your study. We cannot, of course, promise publication at that time. Your revised manuscript is also likely to be sent to reviewers for further evaluation.

We would like to invite the resubmission of a significantly-revised version that takes into account the reviewers' comments. Your revision should address the specific points made by each reviewer, including. 

Sincerely,

Alessandra Aparecida Guarneri

Associate Editor

Paulo Pimenta

Deputy Editor

Reviewer's Responses to Questions

**Key Review Criteria Required for Acceptance?**

**Methods**

-Are the objectives of the study clearly articulated with a clear testable hypothesis stated?

-Is the study design appropriate to address the stated objectives?

-Is the population clearly described and appropriate for the hypothesis being tested?

-Is the sample size sufficient to ensure adequate power to address the hypothesis being tested?

-Were correct statistical analysis used to support conclusions?

-Are there concerns about ethical or regulatory requirements being met?

Reviewer #1: The objectives of the study are clearly stated and the study design is appropriate. Sample sizes are generally adequated. However, concerning Fig 2B a non-parametric test may be more adequate given that n=3.

In Methods section basiconic sensilla are mentioned. However, later in the MS they seem to be called thin-walled trichoid. The nomenclature used should be clarified, citing previous papers and indicating the different names previously given to the same sensillum type (this is to allow the reader compare the information from this and other papers on the same sensillum type). I would suggest using the nomenclature By Shanbag, et al (1999) International Journal of Insect Morphology and Embryology 28 (1999) 377-397, which is in general agreement with Keil (1999) Insect Olfaction (Ed BS Hansson) and Altner, H., Prillinger, L., 1980. International Review of Cytology 67, 69-139, and used in triatomines in, for example, Ref 27 of the MS.

Pls check the formula around line 195, especially in line 201

Reviewer #2: The objectives are clearly articulated and the study design is appropiate to address the stated objetives.

**Results**

-Does the analysis presented match the analysis plan?

-Are the results clearly and completely presented?

-Are the figures (Tables, Images) of sufficient quality for clarity?

Reviewer #1: In Fig 3 I did not undersatnd which responses were significantly different from the negative control. 

Responses to CO2 have been found but nothing is said about those responses.

The resolution of figures 4, 5 should be improved. They are difficult to read.

Reviewer #2: The reading in this section was difficult. Probably because the Figures are quite messed up in the text, which makes reading tough. Therefore, I suggest (as a possibility) to group data differently. For example, data of figure 4A and those of figures 6C and D could be on the same template, where the averaged responses are shown (without function subtype discrimination). And, graphs of Figure 4B could be included along with data of Figure 6A-B. If so, this would need rewriting the text.

**Conclusions**

-Are the conclusions supported by the data presented?

-Are the limitations of analysis clearly described?

-Do the authors discuss how these data can be helpful to advance our understanding of the topic under study?

-Is public health relevance addressed?

Reviewer #1: The conclusions are supported by the data presented.

Reviewer #2: The authors could certainly further develop the limitations of the study and give alternative hypotheses to those provided (which I detail below).

**Editorial and Data Presentation Modifications?**

Reviewer #1: In line 264 the following reference should be added: May-Concha, et al 2016. Infec. Genet. Evol. 40, 73–79

In line 100 (or 106 or 104) the following reference should be added: Guidobaldi & Guerenstein 2016 Journal of medical entomology 53, 770-775

There is a mistake in line 302. Aldehydes have also been tested (see your Ref 27). 

Check line 122 , it seems to contradict a previous sentence.

Line 59 Replace "its" for "a"

Correct TrYpanosoma in Author summary

Reviewer #2: (No Response)

**Summary and General Comments**

Reviewer #1: This MS deals with a comparison of the morphology and physiology of the olfactory system of two triatomine species, one considered domestic, the other sylvatic. A higher density of two types of olfactory sensilla in the sylvatic species was found compared to the domestic one. Also, the sylvatic species presented overall higher and broader olfactory responses than its domestic relative. ( It is suggested that domestic species present a decreased olfactory function, in relation to the limited relevance of this sensorial input in their particular environment with limited olfactory cues. )

In addition, knowledge about the olfactory responses on the antenna of triatomines has been extended, and this includes responses to "new" compounds and a better understanding of functional subtypes. 

The morphology work is adequate and the experiments are correctly carried out. It would be interesting to continue this work , for example, by studying the dose-response characteristic of the ORNs studied. Also, an antennal response to CO2 is reported for the first time in triatomines. It would be interesting to study this further.

In conclusion, this MS contributes new information on triatomines and in general add up to our knowledge on the olfatory system of hematophagous insects.

Reviewer #2: In the manuscript "Comparative dissection of the peripheral olfactory system of the Chagas disease vectors Rhodnius prolixus and Rhodnius brethesi", the authors analyze morphological and physiological differences in the olfactory antennal sensilla between R. prolixus and R. brethesi. They found differences across species in the number of certain morphological types of olfactory sensilla (i.e. thin- and thick- walled trichoid sensilla) and also differences in the olfactory response profile of grooved peg (GP) sensilla. Whereas GP sensilla were found in similar numbers in both species, they showed different olfactory tuning responses. The results here provided are solid, novel and are a useful contribution to the sensory physiology of triatomine insects. 

However, I have major concerns about the construction of the hypothesis and the hypothesis of this work. First, the authors claim that R. prolixus is a domiciliated species, which is not accurate. Whereas R. brethesi until now was only found in sylvatic environments, R. prolixus depending on the region can be found in sylvatic (living in palm trees, e.g. Colombia, Ecuador, Brazil, etc.) or in domestic environments (living in human dwellings, e.g. Guatemala). Therefore, R. prolixus does not represent the best example of an exclusively domiciliated species. An interesting comparison would be to analyze the antennal morphology and physiology of domiciliated and sylvatic R. prolixus. Second, the authors postulate that sylvatic species (R. brethesis) have higher number of sensilla and a larger olfactory tuning of sensory neurons than domicialiated species (R. prolixus) due to their need for an active search for hosts, a requirement that would be higher in sylvatic species than in domiciliated ones. This statement, on which the general hypothesis of the work is based, is weak. In fact, we know that sylvatic species also live in close association with their hosts, probably spending their whole lifecycle in the same vertebrate nest. In these nests they form large colonies and several cohorts coexist. Therefore, asserting that sylvatic triatomines explore and search more actively than domiciliated species, and consequently their need for a more complex sensory machinery than domiciliated ones, is not a solid argument or hypothesis for triatomine species as it might be for other insect species. The differences found with respect to the sensory tools could simply be attributed to species-specific differences related to different host preferences. 

Lastly, reading was difficult. Probably because the Figures are quite messed up in the text, which makes reading tough. Therefore, I suggest (as a possibility) to group data differently. For example, data of figure 4A and those of figures 6C and D could be on the same template, where the averaged responses are shown (without function subtype discrimination). And, graphs of Figure 4B could be included along with data of Figure 6A-B. If so, this would need rewriting the text.

Then, I have other concerns that need to be addressed:

-lines 47-48 "American Tripanosomiasis, also known as Chagas disease, is a disease which no one speaks out…." This affirmation sounds weird when the WHO produces annual reports informing to the scientific and non-scientific community about the statistics of the disease (% of people affected, transmission rates, preventive actions and policies, etc..). Chagas disease is probably less known in Europe or Asia, but it is not the case in the Americas. Moreover, the Center for Disease Control and Prevention (CDC) of the United States considered the Chagas disease as 1 of 5 parasitic infections to be targeted as priority for public health (CDC, 2018).

-line 52 "… and to get their vital blood meal, while infecting them at the same time" this is not precise but is confusing and can be misunderstood. If kissing bugs do not defecate during blood intake, there are no chances of infecting the host, given the parasites are only present in feces and not in the salivary glands. Thus, they can take a small blood meal and leave the host without defecating. I suggest editing the text, to avoid confusion and misunderstandings.

-line 66 replace for Chagas disease

-lines 106-107 "…largely unknown" ? SSR have been carried out before this work, in T. infestans (Guerenstein & Guerin 2001 Diehl et al 2003, Taneja & Guerin 1997, Mayer 1968, Bernard 1974, Guerenstein 1999), R. prolixus (Reisenman 2014) 

Several and previous studies to this work have pointed out the role of the antenna and the olfactory in odour detection. In consequence this sentence is not precise. I suggest editing it.

-lines 109 -111 This is not precise either. Like domestic triatomines sylvatic species also live in close association with their hosts. Sylvatic species live in vertebrate nest, where they can spend in the same nest their entire lifecycle, forming large colonies. Therefore, I disagree with the authors in claiming that sylvatic triatomines explore and search for hosts more actively than domiciliated species.

-line 278. You stated that you've identified 2 new types of sensilla: 1- a peg-in-pit sensillum and a type of coeloconic sensillum. Please specify whether it was an eventual finding or not. If not, please specify in how many preparations you found them. 

-lines 294-295: In figure 2, which is the point of the micrographs shown above the bars of density quantification of sensilla? Please refer to them or exclude them.

- It is shown in Fig.3 that you've tested the response to CO2, please detail in Mat & Met how did you offer the CO2 to the insects, and which was the concentration offered? 

- lines 344-346 : You stated "While only 37% of the odor-sensillum combinations in R. prolixus yielded responses >15 spikes s-1 above solvent response, all combinations did in R. brethesi (Fig. 4, Fig. 5). " However this is not easily to visualize, on the contrary, it looks the opposite. To facilitate cross-species comparisons, I strongly suggest, in Figure 5, putting the Y axes on the same scale.

-line 283. Instead of FigS2E wouldn't it be replace for FigS1E?

-lines 492 - 494 You stated "However, it is conceivable that these chemicals are detected by organs other than the antenna, as the odorant co-receptor orco is expressed also in tarsi, genitalia and rosti". No olfactory sensilla have been identified in tarsi, genitalia or rostrum of triatomine bugs. The presence of odorant-related receptors in these appendages may not necessarily be related to an olfactory function. Please rephrase.

-lines 557- 559 You stated "It seems conceivable that the sylvatic R. brethesi uses olfaction to discriminate between a higher number of hosts and nest sites. In contrast, the domestic R. prolixus might encounter a rather limited number of olfactory stimuli."

I would like to remark that R. brethesi infests only one type of palm tree ( Leopoldinia piassaba ). Consequently, this may also suggest that this wild triatomine has special characteristics in its sensory system allowing specificity for this ecotope.

lines 571- 572 You stated "… the condition found in the sylvatic species represents the ancestral character state in the subfamily, whereas a derived reduced condition is linked with a more or less close association with humans."

It would be interesting to compare morphological and physiological differences between domiciliated and sylvatic R. prolixus.

PLOS authors have the option to publish the peer review history of their article (what does this mean?). If published, this will include your full peer review and any attached files.

Reviewer #1: No

Reviewer #2: No
---

## [Decision Letter · Decision Letter 1]

24 Mar 2021

Dear Dr Silke Sachse,

We are pleased to inform you that your manuscript 'Comparative dissection of the peripheral olfactory system of the Chagas disease vectors Rhodnius prolixus and Rhodnius brethesi' has been provisionally accepted for publication in PLOS Neglected Tropical Diseases.

Best regards,

Alessandra Aparecida Guarneri

Associate Editor

Paulo Pimenta

Deputy Editor

Reviewer's Responses to Questions

**Key Review Criteria Required for Acceptance?**

**Methods**

-Are the objectives of the study clearly articulated with a clear testable hypothesis stated?

-Is the study design appropriate to address the stated objectives?

-Is the population clearly described and appropriate for the hypothesis being tested?

-Is the sample size sufficient to ensure adequate power to address the hypothesis being tested?

-Were correct statistical analysis used to support conclusions?

-Are there concerns about ethical or regulatory requirements being met?

Reviewer #1: The authors have appropriately addressed the concerns of this reviewer regarding this section.

Reviewer #2: The objectives, hypothesis, sample sizes are ok

**Results**

-Does the analysis presented match the analysis plan?

-Are the results clearly and completely presented?

-Are the figures (Tables, Images) of sufficient quality for clarity?

Reviewer #1: The authors have appropriately addressed the concerns of this reviewer regarding this section.

Reviewer #2: Analysis, results and figures are ok in the new version

**Conclusions**

-Are the conclusions supported by the data presented?

-Are the limitations of analysis clearly described?

-Do the authors discuss how these data can be helpful to advance our understanding of the topic under study?

-Is public health relevance addressed?

Reviewer #1: The authors have appropriately addressed the concerns of this reviewer regarding this section.

Regarding one of the author's responses to the comments, I would suggest looking for a response to CO2 among basiconic (single-walled, thin-walled) sensilla.

Reviewer #2: The new version of the manuscript was considerably improved in this matter

**Editorial and Data Presentation Modifications?**

Reviewer #1: I have no further suggestions.

Reviewer #2: The authors have addressed all questions and remarks adequately

**Summary and General Comments**

Reviewer #1: The authors have appropriately addressed the concerns of this reviewer. This version of the MS is highly improved.

Reviewer #2: The authors have addressed all questions and remarks adequately

PLOS authors have the option to publish the peer review history of their article (what does this mean?). If published, this will include your full peer review and any attached files.

Reviewer #1: **Yes: **Pablo Guerenstein

Reviewer #2: No

---

## [Editor Report · Acceptance letter]

8 Apr 2021

Dear Dr. Sachse,

We are delighted to inform you that your manuscript, "Comparative dissection of the peripheral olfactory system of the Chagas disease vectors *Rhodnius prolixus* and *Rhodnius brethesi*," has been formally accepted for publication in PLOS Neglected Tropical Diseases.

Best regards,

Shaden Kamhawi

co-Editor-in-Chief

Paul Brindley

co-Editor-in-Chief
